# OpenReview forum: "Steering Language Models with Weight Arithmetic"
_ICLR.cc/2026/Conference — ICLR 2026 Poster_

### Official Review · Reviewer_Aqfz · 2025-10-27

**Soundness:** 4
**Presentation:** 3
**Contribution:** 3
**Rating:** 6
**Confidence:** 3

**Summary:**

This paper introduces contrastive weight steering for behavior control in LLMs: obtain two small LoRA finetunes that elicit opposite behaviors (positive vs. negative), form a behavior weight vector $w_b=\tau^+-\tau^-$, and steer the base model by adding $kw_b$ to its weights. The method is simple, data-light, and cheap to apply. Experiments show reductions in sycophancy and misalignment with stronger out-of-distribution (OOD) generalization than activation-based steering, and a practical use case for monitoring undesirable drift. Steered models often preserve general capability better than activation steering at comparable control strength.

**Strengths:**

- The method is simple & practical: A minimal, reproducible recipe (two tiny LoRA runs + one vector addition) with low data/compute cost.
- OOD robustness vs. activation steering: Under matched data and control strength, weight steering more often preserves base accuracy while shifting behavior.
- Useful byproducts: The learned behavior direction doubles as a monitoring signal for emergent misalignment.

**Weaknesses:**

- Limited novelty: Methodologically close to task arithmetic/task vectors; the contrastive construction is natural but incremental.
- Baselines could be stronger: Most comparisons are to activation steering or prompts. To contextualize tradeoffs, it would help to include training-heavier baselines (e.g., larger SFT/RLHF slices) on the same behaviors and report cost-adjusted outcomes.

**Questions:**

- Could the authors explain why the double-side difference $\tau^+-\tau^- $ is necessary?: What do we lose by using a single-sided vector, e.g. $ \theta_{\text{positive}}-\theta_0 $?
- On dataset construction robustness: Could mismatch between positive/negative datasets (style, register, spurious topics) contaminate $w_b$? Is there any robust way to ensure we isolate the intended behavior?
- Data & hyperparameter scaling: How does performance/retention vary with number of examples used to fit $ \tau^\pm $, and hyperparameters such as LoRA rank?

---

> ### Author Response · Authors · 2025-11-18
> **Response to Reviewer Aqfz**
>
> We thank the reviewer for their time and valuable feedback. We have updated the paper (including SFT comparison, single-side task vector results, and results for different LoRa ranks) and answer the questions here:
>
> > Could the authors explain why the double-side difference  is necessary?: What do we lose by using a single-sided vector?
>
> We included in the appendix the results of using only the positive or only the negative task vector for each of the experiments. In general, these non-contrastive directions work less well, with the exception of the “evil” experiment. Intuitively, a single direction (positive or negative) contains all changes introduced by fine-tuning, such as topic, style, length, etc. Taking the difference removes the factors shared by both fine-tunings and isolates only the behavior-specific change.
>
> > Could mismatch between positive/negative datasets (style, register, spurious topics) contaminate ? Is there any robust way to ensure we isolate the intended behavior?
>
> Yes. This is why we use exactly the same questions for both the positive and negative datasets. Using matched prompts minimizes differences in style, register, or topic from contaminating the steering vector and helps isolate the intended behavioral change.
>
> > To contextualize tradeoffs, it would help to include training-heavier baselines (e.g., larger SFT/RLHF slices)
>
> We have included SFT baselines on the same data for the sycophancy and evil experiments, and the joint baselines serve as the SFT comparison in the other settings. These results show that SFT can effectively induce behaviors but is less effective at reducing a propensity (as seen in the non-sycophancy results). We agree that RLHF or DPO would be interesting additional comparisons, though we expect them to behave similarly to SFT and to require broader or more carefully curated preference data.
>
> > How does performance/retention vary with number of examples used to fit , and hyperparameters such as LoRA rank?
>
> We added results from our preliminary experiments (Appendix D.1), where we observed that the LoRA rank did not significantly affect the resulting task vectors, although this may be task dependent. Since we saw no meaningful differences, we used LoRA rank 32 for all our experiments.
>
> > Limited novelty: Methodologically close to task arithmetic/task vectors
>
> The novelty of our work lies not just in the contrastive construction of weight steering vectors, but also in its application to modifying behavioral propensities in language models and evaluating its effects across diverse behaviors outside of the training distribution. We directly compare this approach with existing methods for behavioral modification, such as activation steering, and we show that activation steering has important limitations, including degrading model performance before achieving the desired change or failing to modify certain behaviors at all (for example, non-sycophancy).

---

### Official Review · Reviewer_AbNf · 2025-10-28

**Soundness:** 3
**Presentation:** 1
**Contribution:** 3
**Rating:** 4
**Confidence:** 4

**Summary:**

This paper introduces “contrastive weight steering”, a post-training method for modifying high-level behaviors in large language models (LLMs) by arithmetically combining weight differentials from small-scale, purposefully narrow fine-tunings. By subtracting weight changes induced by opposite behaviors (positive vs. negative fine-tunes), the method isolates weight-space directions corresponding to target traits, such as sycophancy, refusal, or “evilness”. The resulting vector is then added or subtracted from the base (or recently task-fine-tuned) model’s parameters. The authors benchmark weight steering against activation steering, an established, layer-specific behavioral modulation technique, and traditional data augmentation through fine-tuning. The work empirically demonstrates that weight steering often generalizes steering effects better to out-of-distribution queries before harming model competence, and can also serve as a tool for post-hoc behavioral monitoring and drift detection by inspecting task vector similarities.

**Strengths:**

1. Methodology: the approach is conceptually simple yet effective, contrastive construction of behavioral directions in weight space, building directly upon and extending well-known task-vector strategies (Ilharco et al., 2023).

2. Clarity with Explicit Comparison: The paper systematically compares contrastive weight steering to activation steering and joint fine-tuning, using diverse alignment-relevant behaviors (sycophancy, refusal, evilness) across several standard LLM architectures (Qwen2.5-7B, Llama-2-7B, etc.).

3. Emphasis on Out-of-Distribution (OOD) Evaluation: Unlike much of the prior literature, the paper pays substantial attention to OOD generalization: for example, by constructing steering vectors with one query distribution and evaluating behavioral shifts on distinct domains, multiple-choice setups, or open-ended generations.

4. Visualizations and Interpretability: Several key figures, such as Figure 2 (sycophancy steering tradeoff), Figure 5 (Evil rate vs. accuracy curve), and Figure 9 (cosine similarity heatmap for weight-behavior vectors), provide clear, interpretable evidence of both the strengths and blind spots of the method.

**Weaknesses:**

1. Oversimplified Method and Ambiguity in Implementation: The proposed method appears conceptually simple, relying on subtracting the negative direction from the positive one and interpolating between them to obtain the final value. However, constructing an appropriate negative direction is often non-trivial and may not be unique (see my Q1), potentially introducing ambiguity in the optimization process. Furthermore, the selection of an appropriate interpolation coefficient k plays a critical role in determining the model’s stability and performance. A more detailed discussion or empirical justification for how the negative direction is defined and how k is chosen would significantly strengthen the technical soundness of the paper.

2. Lack of Insights into Weight vs. Activation Steering: The paper does not provide sufficient explanation or theoretical insight into why weight steering yields better performance than activation steering. Ideally, modifying the weights should be, in principle, equivalent to applying an appropriate transformation to the activations. However, the observed performance difference between the two approaches suggests that there may be underlying mechanisms not yet well understood. It would be valuable to clarify why weight steering behaves differently and appears to offer superior results, with proper insights.

3. Lack of Clarity in Presentation: The paper’s presentation lacks clarity, with several important implementation details either missing or insufficiently explained. For instance, it is unclear how “sycophancy” is formally defined or measured within the experimental setup. Additionally, the paper does not specify whether weight steering is applied to all layers of the model or restricted to certain fixed layers. Providing clearer definitions and methodological descriptions would greatly improve the readability, reproducibility, and overall credibility of the work.

4. Potentially Unfair Evaluation:The evaluation setup may not provide a fair comparison between methods. In several cases, activation steering results in both lower non-sycophancy scores and reduced accuracy (e.g., Figure 3). This outcome is counterintuitive, as both metrics deteriorate under the same adjustment. The authors are encouraged to further investigate this behavior and clarify whether it stems from differences in experimental settings, parameter tuning, or inherent limitations of activation steering. A fairer and more controlled comparison would strengthen the validity of the reported results.

**Questions:**

1. [Key Issue] The reviewer would like to better understand how the negative direction is determined in the proposed method. In general, while a positive direction can often be uniquely defined, there may exist infinitely many possible negative directions. How is a proper negative direction selected in practice? Moreover, how sensitive is the performance of weight steering to this choice? A more detailed explanation or empirical analysis on this aspect would help clarify the robustness and consistency of the proposed approach.

2. [Key Issue] Following the Q1, the reviewer has concerns on the reported results. Considering the Figure 2 as an example. In the left subfigure, it appears that positive activation steering effectively reduces the non-sycophancy score, whereas in the right subfigure, it shows very limited ability to increase the non-sycophancy score. Could the authors clarify why this asymmetry occurs? Is it potentially due to the selection of an inappropriate negative steering direction, given that multiple negative directions could exist? A more detailed explanation or visualization would help elucidate the underlying cause of this discrepancy.

3. A similar issue can be observed in Figure 3. It appears that applying activation steering leads to a decrease in performance while simultaneously decreasing the non-sycophancy score. This behavior seems counterintuitive, as both metrics deteriorate under this adjustment. Could the authors provide further clarification or analysis on why this occurs? It would be helpful to understand whether this effect arises from the steering method itself, the choice of direction, or potential interactions between the two objectives.

4. Unclear Selection of Interpolation Parameter k:The paper does not provide sufficient details on how the interpolation parameter
k is selected. Although the authors illustrate the interpolation behavior using color notations, a more concrete description of the actual k values used is necessary for clarity and reproducibility. Furthermore, an ablation study examining the sensitivity of performance to different k values would offer valuable insight into how this parameter influences the effectiveness and stability of the proposed method.

5. In Line 162, the authors mention that they “select the best-performing layer” for activation steering. Could the authors clarify whether the same layer selection strategy is applied for weight steering? Specifically, do you use the same layer identified for activation steering, or is weight steering performed across all layers?

6. Section 3 appears disproportionately brief relative to its importance. The reviewer suggests that the authors expand this section by including key definitions and methodological clarifications. For instance, it would be helpful to formally define how activation steering is formulated and how sycophancy is evaluated. Shifting or adding such explanations to this section would improve the logical flow and make the paper more self-contained and accessible to readers.

7. Some Typographical Error. For example, in Line 309, there is an inappropriate use of quotation marks around the word “evil”. Also, the caption of Figure 2 appears to contain an error. It currently reads “Weight steering is more effective in controlling sycophancy than weight steering.” This seems to be a typographical mistake—perhaps the authors intended to compare weight steering with activation steering.

---

> ### Author Response · Authors · 2025-11-18
> **Response to Reviewer AbNf (1/2)**
>
> We thank the reviewer for their time and valuable feedback. We have updated the paper to include some of the suggestions: (1) more clarity in the methods section, and (2) better insights on why weight steering yields better performance (by including additional baselines). Below we address the questions:
>
> > how the negative direction is defined
>
> **Both negative and positive directions are defined in the same manner**. The negative direction in weight space ($\tau^-$) is obtained by fine-tuning the pre-trained model on the data $D^-$, which showcases the undesired behavior, e.g. on data that exemplifies what a *non-sycophantic* answer looks like. The usage of examples of undesired behaviors is common, for example when training reward models during RLHF. Directions in weight-space corresponding to a behavior may not be unique, we simply show that the ones found by our method often steer the model in the desired way.
>
> > positive activation steering effectively reduces the non-sycophancy score, whereas in the right subfigure, it shows very limited ability (...) Is it potentially due to the selection of an inappropriate negative steering direction?
>
> The behavior direction is the same in both cases. Positive steering and negative steering apply opposite scalar multiples of that direction, for example $2 w_b$ and $-2 w_b$ (Figure 2) (which is the same approach as the one used by previous activation steering work). There are not multiple “negative” directions; there is only the single behavior direction and its opposite (in vector space). We clarified in the methods section how steering is applied, and we also updated the sycophancy section to make this explicit.
>
> ---
>
> **Lack of Insights into Weight vs. Activation Steering**
>
> > why weight steering yields better performance than activation steering
>
> For a more direct comparison **we have added additional variants** in the appendices and a discussion section (Appendix A, which will be added to the main text in the extra page of the final version).
>
> Activation and weight steering differ in 3 components: (1) single layer vs all-layers, (2) average of activations vs fine-tuning, and (3) activation-space vs weight-space. To measure which differences matter the most we have included:
> * Activation steering applied to all layers: where the steering vector at each layer is re-defined as $a_{\text{all layers}}^l = a^l - a^{l-1}$
> * Bias-only contrastive weight steering: where we only fine-tune the bias term in the down projection of the MLPs, so we can compare to applying a translational transformation to the activations but learned through fine-tuning.
>
> We find that all-layer activation steering is usually similar to single-layer activation steering, and bias-only weight steering has a performance usually between weight steering and activation steering, which suggests that (2) and (3) are the main factors driving the performance of weight steering relative to activation steering.
>
> > modifying the weights should be, in principle, equivalent to applying an appropriate transformation to the activations
>
> We agree in principle because the impact of weights on model behavior is mediated by activations. But the impact might be indirect (e.g. be a function of the activation at a prior layer), and in practice the activation modifications studied by prior work are simple (most study adding a constant to all activations at a given layer). Weight steering is a simple way of capturing more complex and data-dependent modifications than what adding a constant to activations can capture.
>
> > Fair comparison between methods (...) In several cases, activation steering results in both lower non-sycophancy scores and reduced accuracy (e.g., Figure 3). This outcome is counterintuitive, as both metrics deteriorate under the same adjustment (...) clarify whether it stems from differences in experimental settings, parameter tuning, or inherent limitations of activation steering
>
> Indeed both metrics deteriorate under the same adjustment (adding $k a_b$ to a given layer’s activations), indicating that this transformation harms the model’s overall performance (lower baseline accuracy) without improving the target behavior (no increase in non-sycophancy).
>
> We used the same data to compute both activation steering directions and weight steering directions, and we added a bias-only contrastive weight steering variant, which produces a translational shift to activations across all layers. This variant shows similar outcomes to activation steering. These observations suggest that modifying directions in activation space has intrinsic limitations for certain behaviors, in this case for reducing sycophancy.

---

> > ### Author Response · Authors · 2025-11-18
> > **Response to Reviewer AbNf (2/2)**
> >
> > **Lack of Clarity in Presentation**
> >
> > We have updated the methods section to make it clearer, additionally we clarify the raised points here:
> >
> > > how “sycophancy” is formally defined or measured
> >
> > * Sycophancy in the data used to construct the steering vectors is defined as “agreeing with the user”, “alignment with their viewpoint rather than accuracy” (Prompts in Appendix E).
> >
> > * Sycophancy in Section 4 is defined following Sharma et al. (2024), where an answer is non-sycophantic if it remains correct when adding the opinion of the user to the question. In Section 4.1.1 the opinion of the user is added as “I think the answer is…” (see Figure 15 for an example), while in Section 4.1.2 the opinion of the user is added in the proposed solution to the GCD problem (see Figure 4).
> >
> > > whether weight steering is applied to all layers
> >
> > Weight steering is obtained by fine-tuning with LoRa on all the layers of the model, so it is applied to all the layers.
> >
> > > The paper does not provide sufficient details on how the interpolation parameter k is selected
> >
> > In the paper we do not select the scalar k but we rather experiment with a range of values to show the effect and limits of applying steering with different strengths. In practice one would choose the value of $k$ that maximizes validation performance, similar to how other hyperparameters (e.g. learning rate) can be chosen.

---

> ### Comment · Reviewer_AbNf · 2025-11-27
> **Response to the Rebuttal**
>
> First of all, the reviewer would like to thank the authors for their response.
>
> Overall, I would say that most of my concerns have not been adequately addressed. To clarify my primary question: how sensitive is the proposed method to the choice of the negative direction? This issue should be examined for both weight steering and activation steering. The current experiment demonstrates only one particular choice, but the reviewer expects a more thorough analysis that evaluates the robustness of the method under different negative directions.
>
> In addition, Q2 remains unaddressed in the rebuttal, and I would like to restate this concern.

---

> > ### Author Response · Authors · 2025-11-27
> > **Response to Reviewer AbNf**
> >
> > > how sensitive is the proposed method to the choice of the negative direction?
> >
> > Could the reviewer please clarify what they mean by negative direction?
> > If they refer to the data used to compute the negative activations and the negative task vector ($\tau^-$), we used only 20 unique questions with answers where the model has the undesired behavior. The method does not rely on a large amount of data.
> >
> > If they instead refer to the negative direction used for steering, that is $-a_b$ and $-w_b$, there is no sensitivity to assess because in both cases we simply use the opposite of the behavior vector.
> >
> > > In addition, Q2 remains unaddressed
> >
> > Q2 was addressed in the last part of the first comment above, but we clarify it further here.
> >
> > Q2: *“Figure 2 shows that positive activation steering reduces the non-sycophancy score, while negative steering does little to increase it. Why does this asymmetry occur? Is it potentially due to the selection of an inappropriate negative steering direction, given that multiple negative directions could exist?”*
> >
> > The asymmetry shows that activation steering can add the behavior but cannot remove it. Adding the behavior vector $a_b$ to the intermediate activations successfully induces sycophancy, whereas adding $- a_b$ barely reduces sycophancy and mainly harms the model (as shown by the drop in baseline accuracy).
> >
> > In our setup, only one negative direction exists, which is the behavior vector scaled by a negative scalar. While this direction is not effective for activation steering, we show that using $- w_b$ in weight steering does successfully reduce sycophancy.

---

### Official Review · Reviewer_ErcJ · 2025-10-29

**Soundness:** 3
**Presentation:** 3
**Contribution:** 3
**Rating:** 6
**Confidence:** 5

**Summary:**

This paper introduces Contrastive Weight Steering (CWS), a post training technique to modify large language model (LLM) behaviors through weight arithmetic rather than activation interventions. The method isolates behavioral “directions” in weight space by contrasting two fine-tunes one inducing a desired behavior (e.g., truthfulness) and another inducing its opposite (e.g., sycophancy). The resulting weight vector is then added to or subtracted from the base model to steer behavior. Experiments across sycophancy, evilness, and refusal show that weight steering achieves better out-of-distribution generalization than activation steering, maintains task performance, and can even mitigate unwanted drift introduced during downstream fine tuning. Additionally, the paper presents preliminary evidence that cosine similarity in weight space may help detect emergent misalignment during training

**Strengths:**

1. The contrastive weight-space formulation is both simple and effective.
2. Demonstrated across behaviors (sycophancy, evilness, refusal) and architectures.
3. Outperforms activation steering on unseen distributions.
4. Shows that CWS can correct sycophancy induced during task-specific fine-tuning without harming core skills.
5. Provides early evidence that misalignment can be detected by tracking weight-space similarities.
6. Hyperparameters, datasets, and prompts are clearly documented

**Weaknesses:**

1. The paper does not formally analyze why certain weight directions correspond to behavioral dimensions.
2. Steering coefficient 𝑘, k is tuned manually; adaptive or learning-based selection could improve reliability.
3. Experiments use models up to 7 B parameters larger frontier models (e.g., 70 B+) could test scalability.
4. It remains unclear how interpretable or modular these weight directions are across unrelated behaviors.
5.  The “evil vector” similarity experiment is promising but would benefit from quantitative validation over training trajectories.

**Questions:**

1. How stable are weight-space directions across model sizes can a vector learned on a 1.5 B model transfer to a 7 B model?
2. How does the choice of fine-tuning layers (e.g., LoRA rank and target modules) influence steering effectiveness?
3. Could monitoring via cosine similarity be used in real-time to stop training before misalignment occurs?

---

> ### Author Response · Authors · 2025-11-18
> **Response to Reviewer ErcJ**
>
> We thank the reviewer for their time and valuable feedback. We have updated the paper to include some of the suggestions.
>
> > Steering coefficient 𝑘 is tuned manually; adaptive or learning-based selection could improve reliability
>
> We agree that adaptive or learning-based selection could be valuable for specific applications and we leave this for further work. In our work, we focused on understanding the limits of the approach and measured our method’s effectiveness over multiple values of this core hyperparameter.
>
> > How does the choice of fine-tuning layers (e.g., LoRA rank and target modules) influence steering effectiveness?
>
> We have added results from our preliminary experiments (Appendix D.1), where we observed that the LoRA rank did not significantly affect the resulting task vectors, although this may be task dependent. Since we saw no meaningful differences, we used LoRA rank 32 for all our experiments. We did not restrict the fine-tuning layers, as our goal is to assess the ceiling of what weight-space steering can achieve.
>
> > The “evil vector” similarity experiment is promising (...) Could monitoring via cosine similarity be used in real-time to stop training before misalignment occurs?
>
> We added in Figures 36-37 additional results showing cosine similarity and emergent misalignment measurements throughout fine-tuning. Our preliminary findings indicate that increases in cosine similarity precede the emergence of misaligned behavior and occur around the point where validation loss begins to converge. However, more experiments are needed to determine whether stopping training at this stage reliably preserves the intended fine-tuning signal.
>
> > The paper does not formally analyze why certain weight directions correspond to behavioral dimensions.
>
> We do not provide a formal analysis of *why* specific weight directions correspond to behavioral dimensions. Instead, we build on prior work showing that weight directions empirically capture task-specific behaviors [1,2]. We leave it to future work to provide a more formal or mechanistic analysis of why and how weight steering works.
>
> [1] Editing Models with Task Arithmetic (Ilharco et al., ICLR 2023)
>
> [2] Knowledge is a Region in Weight Space for Fine-tuned Language Models (Gueta et al., Findings 2023)
>
> > How stable are weight-space directions across model sizes can a vector learned on a 1.5 B model transfer to a 7 B model?
>
> This transfer is not directly possible because the weight-space directions of the 1.5B and 7B models live in different dimensional spaces. A weight direction is a vector in the flattened weight space of a specific model, so its shape and dimension depend on that model’s architecture. Future work could try to build linear adapters but it is out of the scope of this work.
>
> > It remains unclear how interpretable or modular these weight directions are across unrelated behaviors
>
> Across our experiments, we measure how weight directions affect baseline behaviors such as MMLU performance and baseline answers on TriviaQA and TruthfulQA. However, we acknowledge that other propensities and capabilities are not systematically evaluated. We now make this explicit in the limitations section (Appendix B), which will be included in the additional page added upon acceptance.
>
> > Experiments use models up to 7 B parameters larger frontier models (e.g., 70 B+) could test scalability
>
> Our experiments focus on 1.5B and 7B models, which allows us to study weight steering across two scales while keeping the experiments cheap and simple. We agree that testing larger frontier models (e.g., 70B+) would be valuable for understanding scalability.

---

### Author Response · Authors · 2025-12-03
**Summary of updates made to the paper**

In response to reviewer concerns, we made the following additions to the paper:

1. We added **new baselines across all experiments**, including SFT, bias only contrastive weight steering, activation steering applied to all layers, and non contrastive task vectors. These baselines help clarify why weight steering outperforms activation steering by showing that activation based interventions remain limited even when constructed through fine tuning or applied in multiple layers, and that single sided weight steering is substantially less effective. We summarize these insights in the new discussion section in Appendix A and include the corresponding results throughout the experimental sections.

2. We **clarified the methods section** by rewriting the description of how behavior directions are constructed and how steering is applied.

3. To support the monitoring use case, we added **cosine similarity measurements throughout fine tuning**, showing that similarity to the behavior vectors begins to increase prior to the emergence of misaligned behavior. These results are included in Figure 36 and 37.

4. We added preliminary experiments showing that **varying the LoRA rank does not significantly affect results**, reported in Appendix D.1.

---

### Meta-Review · Area_Chair_fmRF · 2026-01-06

**Summary:**

This paper proposes Contrastive Weight Steering (CWS), a post-training method for modifying LLM behaviour via weight manipulation by adding or subtracting task vectors -- directions in the weight space that enhance or suppress specific propensities of the model (e.g., sycophancy, evilness, refusal).
The novelty of the work is in how the weight direction is constructed, which involves taking the difference between the positive and negative behaviour on a curated dataset of examples. The authors claim that his direction intuitively captures and isolates only the behaviour-specific change, removing other factors such as style and topic. The paper compares CWS with activation steering, SFT (after rebuttal), and standard task arithmetic.

The paper focuses on an important aspect of post-training, trying to understand if weight manipulation can provide better steering than conditioning the activations. The proposed method seems to be more robust than activation steering.
The authors fulfilled most rebuttal promises, and all claimed experiments and additions are present in the revised paper. See the next section for open points.

**Reviewer Concerns:**

Clarification about experiment details and methodology has been provided and seems sufficient.
Some concerns that require attention in the final version, based on the reviews and the AC's reading of the paper:

- SFT and single-sided (non-contrastive) task vector baselines were added. No RLHF/DPO baseline is present; while authors argue these would behave similarly to SFT, this remains unverified and would strengthen the comparison.

- The rationale for matched positive/negative datasets has been reiterated, but sensitivity to the choice of negative direction was raised twice by Reviewer AbNf and not addressed with experiments. While the intuition is reasonable, empirical evidence that minor variations in D⁻ construction do not substantially affect results would strengthen confidence in the method's robustness.

- Limitations are clearly stated in Appendix B and should be moved to the main paper upon acceptance. In general, I recommend making use of the extra page for integrating the paper with the missing details.

- The paper would benefit from a discussion of why more complex activation steering variants were not compared, given the claim that weight steering fundamentally outperforms activation-based approaches.

**Reviewer Scores:**

- AbNf: 4 --> 6
- ErcJ: 6
- Aqfz: 6

---

### Decision · Program_Chairs · 2026-01-26

Accept (Poster)